# END-TO-END SINGLE-STEP FLOW MATCHING VIA DIRECT MODELS

## ABSTRACT

We introduce Direct Models, a flow-matching framework that enables single-step generation by learning a direct mapping from initial noise $x_0$ to all intermediate latent states along the generative trajectory. Our method is trained end-to-end and does not rely on multi-stage distillation. It leverages a progressive learning scheme where the mapping from $x_0$ to $x_{t+\delta t}$ is composed as an update from $x_0$ to $x_t$ plus the velocity at time $t$. This formulation allows the model to learn the entire trajectory in a recursive, data-consistent manner while maintaining computational efficiency. Experimentally, we show that Direct Models achieves state-of-the-art sample quality among single-step flow-matching methods.

## 1 INTRODUCTION

Diffusion models and flow matching methods have recently achieved remarkable success across a wide range of applications, including image synthesis (6), audio generation (8), and 3D shape modeling (11). Despite their effectiveness, a significant limitation of these approaches lies in their reliance on iterative sampling or inference procedures, which are computationally expensive and can limit real-time deployment.

To address this bottleneck, recent works have explored distillation techniques to compress multi-step diffusion or flow-matching models into efficient single-step samplers. Notable examples include (14; 18), which rely on a sequential training procedure: first training a high-quality teacher model and then performing a distillation step. This two-phase approach nearly doubles the overall training time compared to methods that train the model directly.

In contrast, this work proposes a novel *end-to-end training* approach that directly learns a *one-step* diffusion-like generative model without relying on teacher models or distillation. Our method offers a more practical and efficient solution for fast sampling, avoiding the overhead inherent in multi-stage training pipelines and results in high-quality samples (see Figure 1).

To address these challenges, we propose a new class of *Direct Models*, a residual-based formulation that enables both single-step sampling and end-to-end training. The key idea is to directly model the full flow map through a time-indexed residual field, allowing us to query any intermediate latent $x_t$ without requiring numerical integration. This direct access to latents motivates the name of our approach.

At the core of our method lies a simple recursive structure: the residual displacement at time $t + \delta t$ is expressed as a combination of the residual at time $t$ and the local velocity at that point. This recursive formulation not only serves as a training objective but also acts as a structural prior, encouraging consistency across time steps while remaining efficient and fully self-supervised.

Our main contributions are as follows:

- We propose Direct Models, a novel direct residual model for one-step flow generation that enables efficient single-step sampling without iterative inference.

- We introduce a recursive training framework, derived straightforwardly from the finite-difference formulation of the flow, which enforces local velocity consistency and enables end-to-end training of the model.

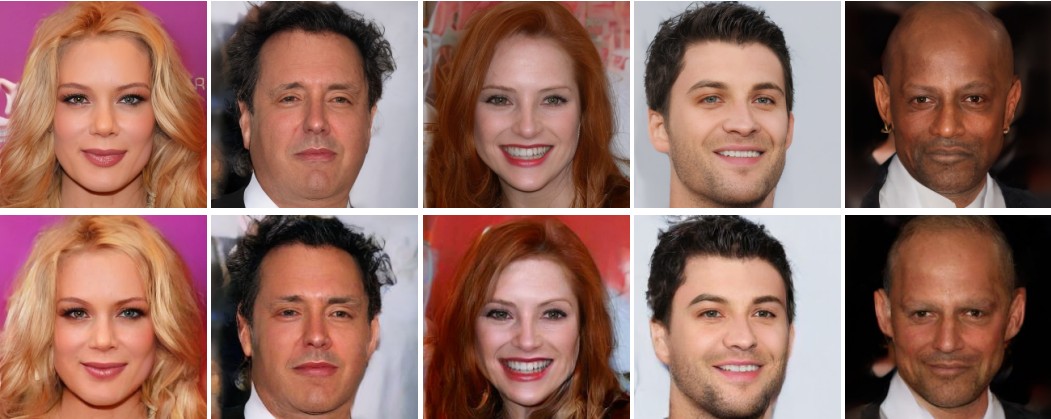

Figure 1: Generations of multi-step flow-matching models and single-step Direct Models. Top row: 128-step generation by a vanilla flow matching model. Bottom row: Generations with our single-step model. Direct Models generates high-quality images across a wide range of inference budgets, including using a single forward pass, drastically reducing sampling time by up to $128\times$ compared to diffusion and flow-matching models. The same starting noise is used within each column.

- We demonstrate that Direct Models achieves superior sample quality compared to existing single-step, end-to-end generative approaches, closing the gap with iterative methods while maintaining fast inference.

## 2 PRELIMINARIES: CONTINUOUS-TIME GENERATIVE MODELS

Modern generative modeling has been significantly shaped by methods that transform simple source distributions into complex data distributions through continuous-time dynamics. Two prominent families in this space are *diffusion-based models* (*e.g.*, (20; 6; 21)) and *flow-based approaches*, particularly those based on *flow matching* (9; 10). These frameworks parameterize sample trajectories using neural differential equations, typically in the form of an ODE, to transport mass smoothly from a source distribution (*e.g.*, Gaussian noise) to a target data distribution.

In this work, we adopt a *flow matching* viewpoint, leveraging its optimal transport-inspired formulation to model deterministic sample paths. Notably, recent studies (*e.g.*, (7)) have emphasized the close relationship between diffusion and flow-based models, observing that flow matching can be viewed as a deterministic instance of more general stochastic diffusion processes. As such, we view these paradigms as conceptually intertwined and refer to them in parallel where appropriate.

Formally, consider a pair of distributions: a base distribution $\mu_0$ and a target distribution $\mu_1$. The goal is to learn a *velocity field* $\mathbf{v}_\theta(x, t)$, parameterized by a neural network, that defines the evolution of a sample over time

$$\frac{d}{dt}\phi(x, t) = \mathbf{v}_\theta(\phi(x, t), t), \text{with } \phi(x, 0) = x_0, \quad x_0 \sim \mu_0. \tag{1}$$

Solving this ODE from $t = 0$ to $t = 1$ generates a trajectory that ideally maps $\mu_0$ into $\mu_1$.

A practical and efficient instantiation of this idea is given by *Conditional Flow Matching (CFM)*, which sidesteps density estimation by using known correspondence pairs $(x_0, x_1) \sim (\mu_0, \mu_1)$. Rather than relying on stochastic score-based gradients, the model is trained to approximate the *ground-truth transport velocity* along straight-line paths

$$x_t = (1 - t)x_0 + tx_1. \tag{2}$$

The true instantaneous velocity along this path is simply $x_1 - x_0$, and the model $\mathbf{v}_\theta(x, t)$ is trained to match this velocity using the loss

$$\mathcal{L}(\theta) = \mathbb{E}_{x_0, x_1, t}\left[\left\|\mathbf{v}_\theta\big((1 - t)x_0 + tx_1, t\big) - (x_1 - x_0)\right\|^2\right]. \tag{3}$$

This supervised objective encourages the model to replicate the optimal displacement between samples at intermediate points in time, by constructing a continuous flow without requiring likelihoods or sampling noise. Once trained, generation consists of drawing $x_0 \sim \mu_0$ and integrating the ODE equation 1 forward using the learned dynamics. This process can be efficiently implemented with standard ODE solvers such as Euler or Runge–Kutta methods.

# 3 METHOD: ONE-STEP FLOW VIA DIRECT MODELS

## 3.1 FORMULATION

We introduce a one-step generative model by directly parameterizing the flow map $\phi(x, t)$. A natural formulation is to define the flow as $\phi(x, t) = x + w(x, t)$, where the residual field $w(x, t)$ represents the displacement from the initial point. However, this formulation allows the displacement magnitude to vary arbitrarily with time, which we found to cause unstable training (see Section 5). To impose a form of temporal consistency, we instead define the flow as

$$\phi(x, t) = x + t \cdot w(x, t), \tag{4}$$

where $w(x, t) \in \mathbb{R}^d$ is now interpreted as a *normalized direction of displacement*, and the scaling by $t$ ensures that the overall displacement grows smoothly from zero to its final value. This parameterization encourages the magnitude $\|t \cdot w(x, t)\|$ to vary linearly with time, providing a stable and interpretable structure for learning. This formulation provides a single-step trajectory, in contrast to the continuous ODE integration approach commonly used in flow matching.

In the flow matching framework, the temporal derivative of the trajectory satisfies

$$\frac{d}{dt}\phi(x, t) = v(\phi(x, t), t), \tag{5}$$

where $v(x_t, t)$ is the target velocity field at the point $x_t = \phi(x, t)$. To incorporate this into our model, we compute the time derivative of $\phi(x, t)$ as defined in Equation equation 4

$$\frac{d}{dt}\phi(x, t) = w(x, t) + t \cdot \frac{\partial w(x, t)}{\partial t}. \tag{6}$$

We approximate the time derivative of $w(x, t)$ using the forward difference with a discrete $\delta t$ step

$$\frac{\partial w(x, t)}{\partial t} \approx \frac{w(x, t + \delta t) - w(x, t)}{\delta t}. \tag{7}$$

By substituting this approximation into the derivative of $\phi(x, t)$, we obtain

$$\frac{d}{dt}\phi(x, t) = w(x, t) + t \cdot \frac{w(x, t + \delta t) - w(x, t)}{\delta t}. \tag{8}$$

By matching this expression to the target velocity $v(x_t, t)$, we then have

$$w(x, t) + t \cdot \frac{w(x, t + \delta t) - w(x, t)}{\delta t} = v(x_t, t), \tag{9}$$

which can be rearranged into the equation

$$t \cdot \frac{w(x, t + \delta t) - w(x, t)}{\delta t} = v(x_t, t) - w(x, t). \tag{10}$$

Finally, by multiplying both sides by $\frac{\delta t}{t}$ we have

$$w(x, t + \delta t) - w(x, t) = \frac{\delta t}{t} \cdot (v(x_t, t) - w(x, t)), \tag{11}$$

and by isolating $w(x, t + \delta t)$, we arrive at

$$w(x, t + \delta t) = \frac{t - \delta t}{t} \cdot w(x, t) + \frac{\delta t}{t} \cdot v(x_t, t). \tag{12}$$

## 3.2 TRAINING DIRECT MODELS VIA LOCAL VELOCITY PROPAGATION

To learn the flow map parameterized by a model $w_\nu$, with parameters $\nu$, we leverage the recursive structure implied by the progressive velocity propagation equation

$$w(x, t + \delta t) = \frac{t - \delta t}{t} \cdot w(x, t) + \frac{\delta t}{t} \cdot v(x_t, t). \qquad (13)$$

This relation connects the residual field $w$ at two consecutive time steps through the velocity field $v$. We exploit this property to define a consistency-based training loss for $w_\nu$, encouraging it to align with the propagated velocity information.

In our formulation, we train two models jointly:

- $v_\theta(x, t)$: a velocity field trained using the standard Conditional Flow Matching (CFM) loss,
- $w_\nu(x, t)$: a residual displacement field trained using a recursive propagation loss derived from Equation equation 13.

The velocity field $v_\theta$ is trained using the standard Conditional Flow Matching (CFM) loss. Given a sample pair $(x_0, x_1) \sim (\mu_0, \mu_1)$ and a uniformly sampled time $t \sim \mathcal{U}[0, 1]$, we define the intermediate point

$$x_t = (1 - t) \cdot x_0 + t \cdot x_1. \qquad (14)$$

The CFM objective encourages the predicted velocity to match the ground-truth displacement between $x_0$ and $x_1$ at this intermediate point

$$\mathcal{L}_{\text{CFM}}(\theta) = \mathbb{E}_{x_0, x_1, t} \left[ \| v_\theta(x_t, t) - (x_1 - x_0) \|^2 \right]. \qquad (15)$$

The residual field $w_\nu$ is trained using a local velocity propagation loss derived from Equation equation 13. Given the sample $x_0 \sim \mu_0$, a small step size $\delta t$ and $t' \sim \mathcal{U}[\delta t, 1 - \delta t]$, we define the propagation loss as

$$\mathcal{L}_{\text{prop}}(\nu) = \mathbb{E}_{x_0, t'} \left[ \left\| w_\nu(x_0, t' + \delta t) - \left( \frac{t' - \delta t}{t'} \cdot \text{sg}[w_\nu(x_0, t')] + \frac{\delta t}{t'} \cdot v_\theta(x_t', t') \right) \right\|^2 \right], \qquad (16)$$

with $x_t' = x_0 + t' \cdot \text{sg}[w_\nu(x_0, t')]$, where $\text{sg}[\cdot]$ denotes a stop-gradient operator. Notice that we define the residual field model $w_\nu$ only with respect to the samples $x_0$ from the initial distribution $\mu_0$. In this way, at inference we can directly map these samples to any point along the trajectory in one step, and, in particular, to the target distribution samples $x_1$. Although $w_\nu(x_0, t')$ may be initially uninformative at the beginning of the training, the propagation loss remains effective, removing the need for explicit scheduling of $t'$. This simplifies training, improving stability and practicality without compromising performance. Importantly, the training of the velocity $v$ is decoupled from the training of $w$, which enables both components to be learned *jointly* and *in parallel*. As a result, the overall training time is approximately equivalent to that of a single generative model.

Our training algorithm is outlined in Algorithm 1.

## 3.3 SAMPLING FROM DIRECT MODELS

Sampling from our direct flow map model is straightforward and efficient. Given an initial sample $x_0 \sim \mu_0$, the corresponding transformed sample $x_1$ can be obtained via a single forward pass of the residual field

$$x_1 = x_0 + w_\nu(x_0, 1). \qquad (17)$$

This one-step sampling eliminates the need for iterative procedures, making the approach practical and fast for inference.

## 4 EXPERIMENTS

### 4.1 SETTINGS

In this section, we compare our method against several existing approaches. All models are trained from scratch using the same architecture and implementation framework to ensure a fair comparison.

---

**Algorithm 1** Training Direct Models via Local Velocity Propagation

---

1: Initialize parameters $\theta$ for $v_\theta$, $\nu$ for $w_\nu$
2: **for** each training step **do**
3:     Sample pair $(x_0, x_1) \sim (\mu_0, \mu_1)$
4:     **Train velocity model $v_\theta$ with CFM loss:**
5:     Sample $t \sim \mathcal{U}[0, 1]$
6:     Compute $x_t = (1 - t)x_0 + tx_1$ and
7:     Minimize

$$\mathcal{L}_{\text{CFM}} = \|v_\theta(x_t, t) - (x_1 - x_0)\|^2$$

    with respect to $\theta$
8:     Update $\theta$
9:     **Train residual field $w_\nu$ with local propagation loss:**
10:     Sample $t' \sim \mathcal{U}[\delta t, 1 - \delta t]$
11:     Compute $x'_t = x_0 + t' \cdot \text{sg}[w_\nu(x_0, t')]$
12:     Minimize

$$\mathcal{L}_{\text{prop}} = \left\| w_\nu(x_0, t' + \delta t) - \left( \frac{t' - \delta t}{t'} \cdot \text{sg}[w_\nu(x_0, t')] + \frac{\delta t}{t'} \cdot v_\theta(x'_t, t') \right) \right\|^2$$

    with respect to $\nu$
13:     Update $\nu$
14: **end for**

---

Specifically, we adopt the DiT-B diffusion transformer architecture from (16). Our experiments include unconditional generation on the CelebAHQ-256 dataset (12) and class-conditional generation on ImageNet-256 (2). For the results reported in Table 1, we use the AdamW optimizer with a fixed learning rate of $5 \times 10^{-5}$ and no weight decay. Additionally, all models operate in the latent space provided by the `sd-vae-ft-mse` autoencoder (17). Further implementation details are provided in Appendix 5. We release the full code in the supplementary materials.

## 4.2 COMPARED METHODS

We compare our method to several prior end-to-end approaches, following the same evaluation setup as in (3). Consistency Training (23) is an end-to-end method that trains a one-step model directly on empirical pairs $(x_t, x_{t+\delta})$, with time discretization bins increasing progressively during training. Extensions such as iCT (22) and sCT (13) build on (23) with modified training optimizations, further improving performance. Shortcut Models (3) propose a generative framework that conditions on both the current noise level and desired step size, enabling efficient and flexible sampling across different inference budgets. Finally, Live Reflow (3) is an end-to-end model trained simultaneously on flow-matching and Reflow-distilled targets. The model is conditioned separately on each type of target, and new distillation targets are generated at every training step via full denoising, making the method computationally expensive.

## 4.3 EVALUATION

We follow the evaluation protocol from (3). Models are evaluated by generating samples using 1 diffusion step for our method, and 128, 4, and 1 steps for the baselines. We report the FID-50k score, a standard metric in generative modeling. FID is computed using statistics from the full dataset, with no compression applied to the generated images. All images are resized to $299 \times 299$ using bilinear interpolation and clipped to the $(-1, 1)$ range. During evaluation, we use the Exponential Moving Average (EMA) of the model parameters.

Table 1: **Comparison of various training objectives applied to the same architecture (DiT-B)**. We report FID-50k scores (lower is better) for 128, 4, and 1-step denoising. Direct Models achieves high-quality samples using a single training phase and a one-step inference process. Results in parentheses indicate settings beyond the intended use of the corresponding objective.

| End-to-End Methods | CelebAHQ-256 | | | ImageNet-256 (Class-Conditional) | | |
|---|---|---|---|---|---|---|
| | **128-Step** | **4-Step** | **1-Step** | **128-Step** | **4-Step** | **1-Step** |
| Diffusion (6) | 23.0 | (123.4) | (132.2) | 39.7 | (464.5) | (467.2) |
| Flow Matching (9) | 7.3 | (63.3) | (280.5) | 17.3 | (108.2) | (324.8) |
| CT (23) | 53.7 | 19.0 | 33.2 | 42.8 | 43.0 | 69.7 |
| iCT (22) | - | - | 21.7 | - | - | 43.3 |
| sCT (13) | - | - | 19.3 | - | - | 41.6 |
| Live Reflow (3) | **6.3** | 27.2 | 43.3 | 46.3 | 95.8 | 58.1 |
| Shortcut Models (3) | 6.9 | **13.8** | 20.5 | **15.5** | **28.3** | 40.3 |
| Direct Models (Ours) | - | - | **14.1** | - | - | **34.4** |

Table 2: Effect of $\delta t$ on image quality on CelebAHQ-256.

| $\delta t$ | 0.005 | 0.01 | 0.05 | 0.1 |
|---|---|---|---|---|
| FID $\downarrow$ | 16.6 | 16.8 | 20.5 | 25.4 |

## 4.4 RESULTS

Direct Models achieves high-quality generation with just a single sampling step. It outperforms all existing single-stage training approaches for one-step generation. Additional qualitative results are presented in Appendix A.

## 5 ABLATION

In Table 2, we study the effect of the finite difference step size $\delta t$ on image quality. We find that smaller values of $\delta t$ (0.005, 0.01) achieve the best FID scores, whereas larger step sizes (0.05, 0.1) result in a clear deterioration in performance. These results highlight the importance of choosing a sufficiently small $\delta t$ to ensure accurate finite-difference approximations and stable training.

In Table 3, we report the FID scores of generated images when training with and without gradient stopping. We observe that disabling gradient stopping results in (i) lower image quality and (ii) approximately 20% additional computational overhead.

In Table 4, we compare the effect of two flow parameterizations during training: $\phi(x, t) = x + w(x, t)$ and $\phi(x, t) = x + tw(x, t)$. Although these two formulations are mathematically equivalent, we observe that training with $\phi(x, t) = x + w(x, t)$ leads to oscillating loss values and consistently produces degenerate images. This observation aligns with the common practice in training diffusion models and flow matching, where a normalized score or velocity is learned rather than the absolute residual.

Table 3: Effect of applying gradient stopping during training (as in Equation 16) on image quality for CelebAHQ-256.

| | w/o stopping gradient | w/ stopping gradient |
|---|---|---|
| FID $\downarrow$ | 42.6 | 16.8 |

Table 4: Effect of flow $\phi(x,t)$ parametrization during training on image quality on CelebAHQ-256.

|  | $\phi(x,t) = x + w(x,t)$ | $\phi(x,t) = x + tw(x,t)$ |
|---|---|---|
| FID $\downarrow$ | > 100 | 16.8 |

## 6 RELATED WORK

We briefly review existing approaches that enable single-step diffusion-based generation, which can be broadly categorized into distillation-based methods and single-phase training methods.

### 6.1 DISTILLATION METHODS

In recent years, various techniques have been developed to distill generative models, particularly diffusion models, into more efficient one-step sampling frameworks. These methods typically follow a two-stage pipeline: **first**, a diffusion model is pretrained; **then**, a separate model is trained to approximate the behavior of the full diffusion process using fewer inference steps.

Methods such as knowledge distillation (14) and rectified flows (10) generate synthetic training pairs by fully simulating the reverse-time denoising ODE. To reduce the high computational cost of full simulation, more efficient alternatives use bootstrapping strategies to partially initialize the ODE trajectory (5; 24). Other works explore alternatives to the standard L2 loss, including adversarial objectives (19) and distribution-matching approaches (26; 25). Progressive distillation methods (18; 1; 15) further break the distillation process into multiple stages with increasing time step sizes, reducing the need for long bootstrap paths.

The distillation method most similar to ours is BOOT (5). Both BOOT and our Direct Models aim to map the initial noise directly to all intermediate latents. However, BOOT is based on a diffusion formulation, while Direct Models adopt a flow-matching formulation, which is an independent class of generative models with distinct derivations and training losses. Importantly, BOOT is a two-stage distillation framework that relies on a pre-trained generative model, whereas Direct Models achieves state-of-the-art performance through a single joint training stage. This demonstrates that joint training is not only feasible but also highly effective, removing the sequential training bottleneck inherent in two-stage distillation methods.

In contrast, we propose an end-to-end approach that directly learns a one-step generative model, removing the need for separate pretraining and distillation phases. This makes our method simpler and more efficient than both full simulation-based techniques and multi-stage progressive distillation.

### 6.2 SINGLE-PHASE TRAINING METHODS

Few methods have been proposed for single-phase training that enable one-step generation. Consistency Models (23), a pioneering approach in this area, directly map partially noised data points to their fully denoised outputs in a single step. While initially developed for distillation, these models have also been explored in end-to-end training scenarios. Extensions such as iCT (22) and sCT (13) build on (23) with modified training optimizations, further improving performance. Shortcut Models (3) propose a novel generative framework that conditions on both the current noise level and the desired step size, enabling efficient and flexible sampling across different inference budgets.

A recent *concurrent work*, Mean Flows (4), introduces a one-step generative modeling approach that captures the average flow along the trajectory. While (4) focuses on learning a direct mapping between every pair of points along the flow, our method targets a direct mapping from the initial noise to all intermediate latent states. Additionally, Mean Flows relies on computationally expensive Jacobian matrix computations, whereas our approach uses a more efficient finite-difference approximation of the flow.

**Difference from Consistency Models (23; 13; 23)** While we share the general concept of consistency with (23; 13; 23), Direct Models differs significantly in its formulation. Conceptually, Direct Models can be seen as the opposite approach: instead of mapping intermediate latents $x_t$ directly to the fully

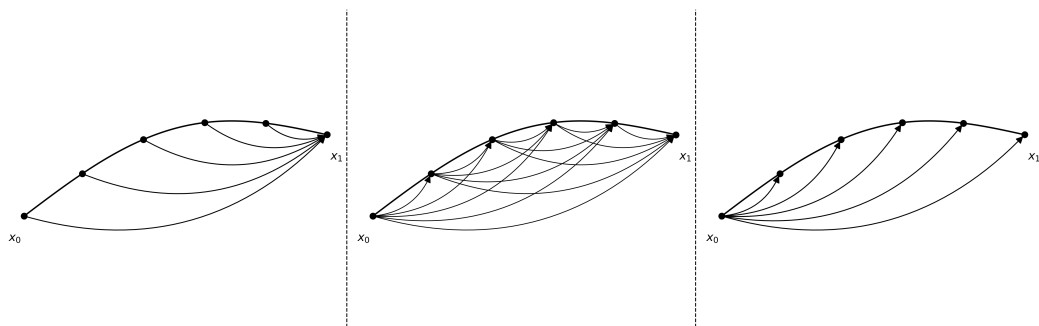

Figure 2: Visual illustration of the differences between prior work in terms of the learned trajectory mappings. $x_0$ denotes an initial Gaussian noise and $x_1$ its corresponding noise-free image. Left: Consistency models (23) . Middle: Shortcuts models (3). Right: Direct Models (ours).

denoised image $x_1$, our method maps initial Gaussian noise $x_0$ to all intermediate latent states $x_t$, as shown in Figure 2.

**Difference from Shortcut Models (3)** Shortcut models are arguably the most similar approach to Direct Models. However, there are key differences: 1) Conceptually, Shortcut models learn direct mappings between all pairs of latent states, including both the initial and final ones. In contrast, Direct Models focuses on learning a direct mapping only between the initial Gaussian noise and all intermediate latent states. 2) More importantly, our approach is grounded on a more principled mathematical formulation. Specifically, we derive our method using finite differences of the flow, as shown in Equation equation 13, which links the flow at neighboring timesteps to the velocity field.

## 7 LIMITATIONS AND FUTURE WORK

Our method has certain limitations. The current formulation is restricted to single-step inference, and Direct Models could be further optimized by training a single network instead of two. A promising direction for future work is to extend the framework to a unified model that allows flexible sampling with a variable number of inference steps.

## 8 REPRODUCIBILITY STATEMENT

To support reproducibility, we provide a complete implementation of our method, including training and evaluation scripts, as part of the supplementary materials.

## 9 CONCLUSION

We introduce Direct Models, a flow matching based generative model that enables both single-step sampling and end-to-end training. By learning a time-indexed residual field to directly approximate the full generative flow, our method achieves fast, high-quality generation. This makes Direct Models a practical alternative to existing single-step generative modeling techniques.

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

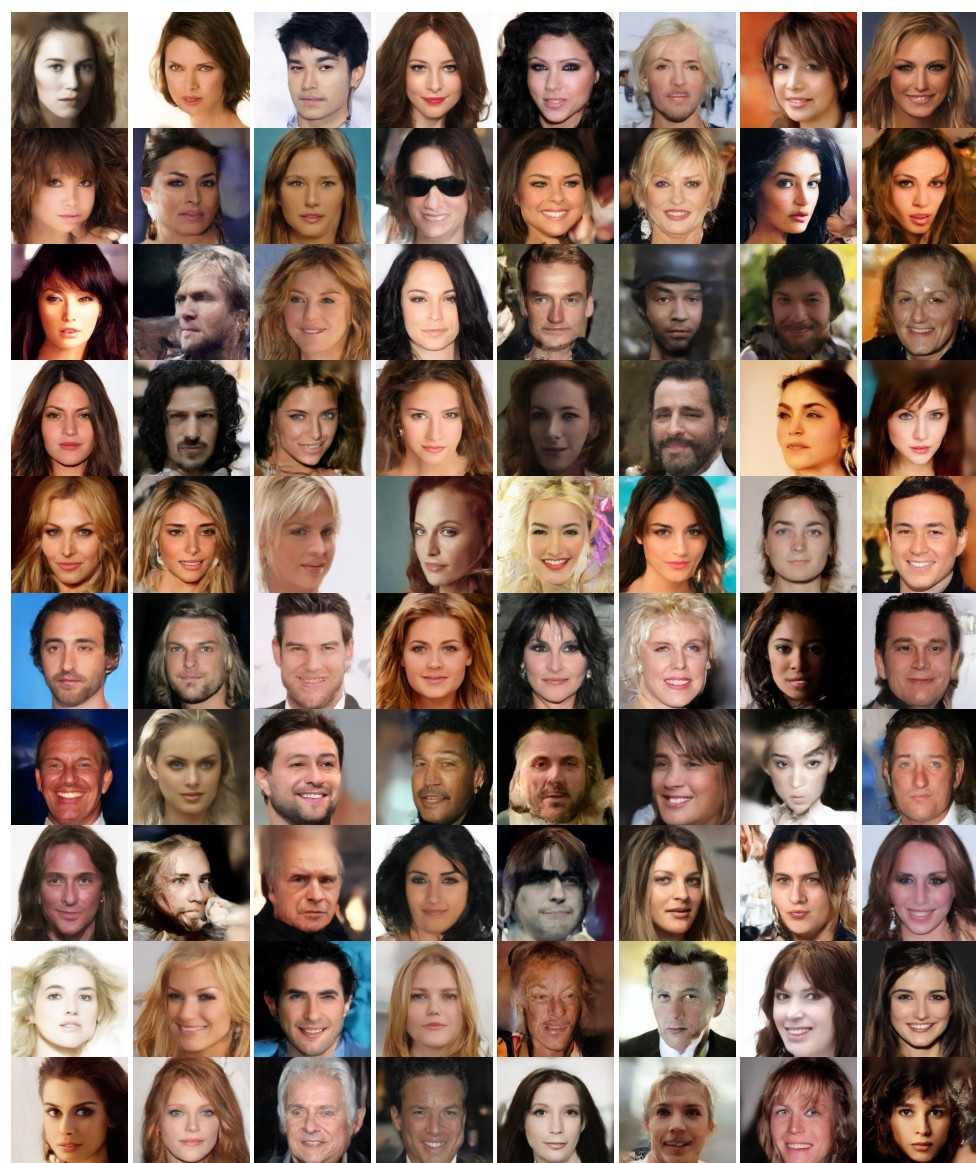

Figure 3: Representative examples generated unconditionally on the CelebA-HQ dataset at 256×256 resolution, using single-step generation with a DiT-B size model trained for 500,000 iterations.

## A  VISUAL RESULTS

Figures 3 and 4 show images generated by our method, trained on the unconditional CelebA-HQ and class-conditioned ImageNet datasets, respectively.

## B  TRAINING DETAILS

Table 5 provides detailed training configurations.

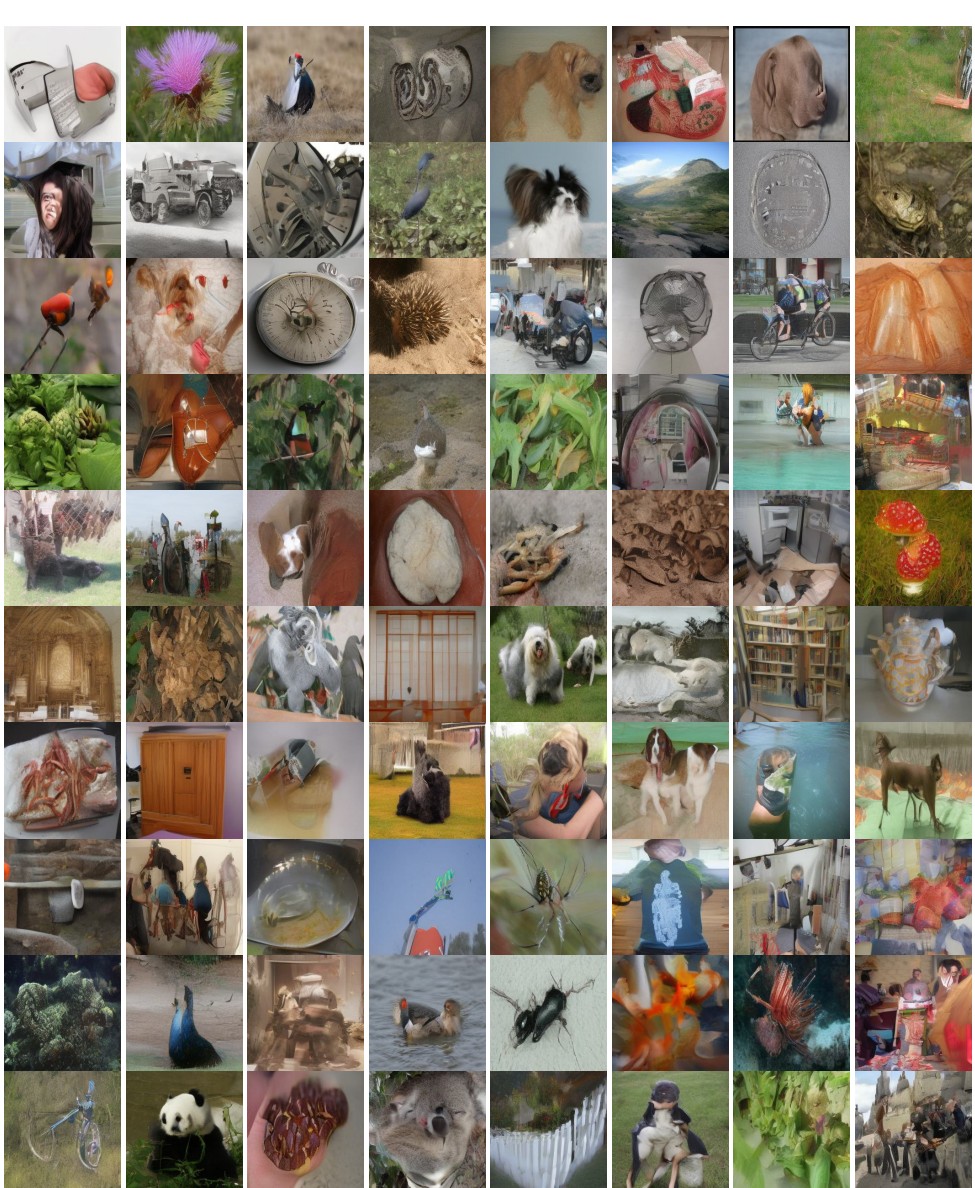

Figure 4: Representative examples generated unconditionally on the ImageNet dataset at 256×256 resolution, using single-step generation with a DiT-B size model trained for 800,000 iterations.

| | |
|---|---|
| Batch Size | 64 (CelebA-HQ), 256 (Imagenet) |
| Training Steps | 500,000 (CelebA-HQ), 800,000 (Imagenet) |
| Latent Encoder | sd-vae-mse-ft |
| Latent Downsampling | 8 (256x256x3 to 32x32x4) |
| Classifier Free Guidance | 0 (CelebA-HQ), 1.5 (Imagenet) |
| Class Dropout Probability | 0 (CelebA-HQ), 0.1 (Imagenet) |
| EMA Parameters Used For Evaluation? | Yes |
| EMA Ratio | 0.999 |
| Optimizer | AdamW |
| Learning Rate | 0.00005 |
| Weight Decay | 0.0 |
| Hidden Size | 768 |
| Patch Size | 2 |
| Number of Layers | 12 |
| Attention Heads | 12 |
| MLP Hidden Size Ratio | 4 |
| $\delta t$ | 0.01 |

Table 5: Hyperparameters used during training.

