# OpenReview forum: "End-to-End Single-Step Flow Matching via Direct Models"
_ICLR.cc/2026/Conference — ICLR 2026 Conference Withdrawn Submission_

### Official Review · Reviewer_i99M · 2025-10-27

**Soundness:** 3
**Presentation:** 3
**Contribution:** 3
**Rating:** 6
**Confidence:** 4

**Summary:**

The authors present a method for learning flow matching (FM) models with single step generation capability. Authors aim to learn the flow map that maps starting point $x_0$ to point $x_t$ from FM generation trajectory that starts at $x_0$. The flow map is parametrized as a residual with a learnable “residual displacement” term that is learned by utilizing the finite difference dynamics of the flow map. In practice, authors modify the learning procedure of the regular flow matching by adding a residual displacement neural network and learning it in parallel with the FM vector field. The authors provide experiments on CelabaHQ and ImageNet datasets and compare with other methods for few step generation of diffusion models and flow matching.

**Strengths:**

- The idea to utilize finite difference for learning the flow map is clean and simple. The loss function is a simple MSE and doesn’t require adversarial optimization.
- Experiments show decent one step generation performance on both CelebaHQ and ImageNet datasets. This is a significant contribution to the acceleration of the FM research field.
- Ablation studies on finite difference step size, various stop gradient strategies in the loss and flow map parametrizations are presented.
- Comparison with other models with the same neural network architecture and in the same latent space.

**Weaknesses:**

- In the experimental section there is no comparison with adversarial methods for one and few step generation from both diffusion and flow matching models, e.g., [1, 2]. The Table 1 in [2] has a lot of methods in comparison, so it would be nice to see something like this.
- The method cannot be used for a few step generation, only for one step generation. This is a serious restriction of the method. While other FM acceleration methods present an ability to perform multistep generation [3, 4]
- Line 376, should (23, 13, 22) be cited instead of (23, 12, 23)?
- MeanFlows [4] is extremely relevant and more than 5 months since the release of their paper on arxiv has passed, so I would like to see the comparison with them as well.
- Seems like in practice your learning procedure would require two times more compute than the regular flow matching. Although it is comparable with the shortcut models [3], still it is a downside. Can you extend your method for learning the flow map in a fine tuning regime (distillation)?
- The method requires learning of two distinct neural networks. It is the case for many acceleration methods, but still it is a downside.
- In the ablation study on CelebaHQ dataset the best model has FID of 16.6, while Table 1 shows FID of 14.1 for the proposed method. Can authors explain this mismatch?
- The paper has too little content. Although itself it is not a problem, because all the necessary details are in place.

[1] Huang, Z., Geng, Z., Luo, W., & Qi, G.-J. (2024). Flow Generator Matching. arXiv preprint arXiv:2410.19310.

[2] Yin, Tianwei; Gharbi, Michaël; Park, Taesung; Zhang, Richard; Shechtman, Eli; Durand, Frédo; Freeman, William T. (2024). Improved Distribution Matching Distillation for Fast Image Synthesis. In Proceedings of the NeurIPS 2024.

[3] Frans, K., Hafner, D., Levine, S., & Abbeel, P. (2025). One Step Diffusion via Shortcut Models. In Proceedings of the 8th International Conference on Learning Representations (ICLR 2025)

[4] Geng, Z., Deng, M., Bai, X., Kolter, J. Z., & He, K. (2025). Mean Flows for One-step Generative Modeling. arXiv preprint arXiv:2505.13447

**Questions:**

- Table 2 shows that the lower $\delta t$ the better. Why don’t authors choose the even lower $\delta t$. In general what would happen if $\delta t$ is too low?

**Details Of Ethics Concerns:**

-

---

### Official Review · Reviewer_H96y · 2025-10-30

**Soundness:** 2
**Presentation:** 1
**Contribution:** 2
**Rating:** 2
**Confidence:** 4

**Summary:**

This paper proposes Direct Models, an end-to-end, single-step flow matching framework. The method parameterizes a flow map $\phi(x, t)=x+tw(x, t)$ with $x\sim \mu_0$, and couples a residual field $w_v$ with a a velocity field $v_{\theta}$ via a finite-difference recurrence. The two networks are trained alternately: $v_\theta$ with a CFM loss and $w_ν$ with a propagation loss. Sampling is a single forward pass. Experiments are conducted on CelebA-HQ 256 and ImageNet 256 for single step generation.

**Strengths:**

+ Practical goal. One step generation without distillation is a promising and meaningful direction.
+ Simplicity. The approach is straightforward and can be implemented atop standard flow matching backbones.

**Weaknesses:**

- Writing quality. The introduction is unfocused and reads like a partial draft. Section 3.1 is overloaded with algebraic manipulations (Eqs. 4–12) and offers little intuition; the derivation could be condensed to a brief statement.
- Limited evaluation. Table 1 compares mainly to earlier one step methods; key contemporaries (e.g., MeanFlow and strong one step distillation baselines) are missing, making the comparison incomplete. It is also important to evaluate across multiple backbones, not only DiT B. Ablations are presented as single numbers without deeper analysis.
- Concerns about training design. Two networks are trained alternately ($v_\theta$ and $w_v$),  with the target for $w_ν$ depending on the continually changing $v_\theta$. This moving target, bi level optimization lacks a stability analysis, so convergence is unclear.
- Weak contribution. Although the paper outlines differences from MeanFlow, the proposed method appears to be a narrower, forward only special case of MeanFlow and lags behind MeanFlow by a large margin. Overall contribution is therefore limited.

**Questions:**

- Can you provide stability/convergence analysis for the alternating optimization of $v_\theta$ and $w_ν$ (Algorithm 1), or empirical evidence (e.g., multi seed curves) that it is stable?
- How does your method compare to MeanFlow on the same setups? Your Related Work mentions MeanFlow; a direct, controlled comparison is needed.
- Please expand Section 3.1 with intuition and move most algebraic steps to the appendix.

---

### Official Review · Reviewer_62yj · 2025-11-01

**Soundness:** 2
**Presentation:** 2
**Contribution:** 1
**Rating:** 4
**Confidence:** 4

**Summary:**

The authors propose a variant of Consistency Model [1], which they called “Direct Models”, to obtain a one-step data sampler trained simultaneously with a Flow Matching model using a consistency loss. The authors demonstrate the performance of their methodology on a number of image generation tasks
[1] Consistency Models, Song et. al., ICML’23

**Strengths:**

- The authors clearly explain the origins of the consistency loss in case of flow-based models

- While their method is essentially similar, it has certain differences with (existing) Consistency models

- Promising practical performance

**Weaknesses:**

To my understanding, the main deficiency of the paper is the lack of novelty. Essentially, the authors propose a method quite similar to Consistency Models [1] (and a number of other works, which develop this idea), particularly, Consistency Distillation (CD) and Shortcut  variant of these models. I found the following differences between CD and proposed “Direct Models”:

- the authors learn flow model and its distilled version *simultaneously*, not one after the other. I think this “simultaneous” framework is strange, because at the initial stages the velocity model $v_{\theta}$ does not provide reasonable information for training the distillation;

- As explained in Sec 6.1., the authors learn a mapping from noise to any trajectory stage, while in the original CT/CD models the mapping is learned from any trajectory stage to the end (data) point. I think, the second variant is more reasonable, because it allows to perform *few-step* sampling for better quality (compared to one-step sampling).

Probably, the outlined differences are more serious than I think, and are components which lead to improved performance. In this case, I recommend authors to study (theoretically/practically) the differences in details, to understand this point.

**Questions:**

No questions

---

### Official Review · Reviewer_pBE4 · 2025-11-01

**Soundness:** 2
**Presentation:** 3
**Contribution:** 2
**Rating:** 2
**Confidence:** 3

**Summary:**

This paper presents Direct Models, a new framework for training one-step generative models within a flow matching framework. The main idea is rather common: to train a direct, time-indexed mapping from the initial noise $x_0$ to all intermediate states of the generative trajectory, which allows for one-step sampling without the need for a separate, computationally expensive distillation process. The method is evaluated on standard image generation tests (CelebA-HQ, ImageNet) and shows competitive FID results compared to other one-step methods.

**Strengths:**

- The central idea of directly modeling the residual field $w(x, t)$ to have single-step access to the entire trajectory is novel. The proposed recursive structure, derived from a finite-difference approximation of the flow ODE (Eq. (12)-(13)), provides a principled and intuitive mechanism for enforcing temporal consistency.

- An advantage of this work is its single-stage training procedure. By decoupling and jointly training the velocity field $v_\theta$ and the residual field $w_\nu$, the method *formally* avoids the two-stage pipeline (teacher training + distillation) that plagues many other single-step methods.

- The experimental results demonstrate good FID scores among single-phase, one-step methods on both CelebA-HQ-256 and ImageNet-256. The fact that it outperforms methods like Consistency Training (CT), improved CT (iCT), and Shortcut Models on the challenging ImageNet-256 benchmark is a strong point in its favor.

**Weaknesses:**

- The main weakness is the lack of comparison with other similar methods. The comparison, although generally good, could be more comprehensive. The results presented in Table 1 are convincing when compared to other end-to-end methods. However, a comparison with the final performance of SOTA distillation-based methods (e.g., Adversarial Diffusion Distillation (ADD) [19], Distribution Matching Distillation (DMD) [26]) in a single-stage mode would provide a more complete picture. Although the argument for training efficiency remains valid, it is critical for the community to know the absolute performance gap. Is the performance of direct models (FID 14.1/34.4) comparable to these distilled models, or is there still a gap that is a trade-off for single-step training?


- The paper trains two models ($v_\theta$ and $w_\nu$) jointly. It claims the total training time is "approximately equivalent to that of a single generative model," which is plausible if they share parameters or have similar sizes. However, this needs clarification. Are $v_\theta$ and $w_\nu$ two separate networks? If so, this doubles the parameter count. If they share an encoder with separate heads, this should be stated explicitly, as it is an important implementation detail affecting the method's efficiency.

- For the same reason, there is a lack of comparison with two-phase methods. The authors claim that both models ($v$ and $w$) are trained in parallel. However, this may require twice as much memory for training, which may not fit within existing hardware requirements. Thus, although formally the training is parallel, in essence, model $v_theta$ is the teacher for $w_\nu$ in the presented work, and therefore comparisons with other two-phase methods or similar distillation methods are necessary.

- The term "Direct Model" is misleading. The model does **not** learn a single direct mapping from $x_0$ to $x_1$; rather, it learns a field $w_\nu(x_0, t)$ that is trained via a consistency loss across time steps (Eq. 16). This is more analogous to a consistency model trained from $x_0$ rather than $x_t$. The description in the abstract and introduction could be refined to more precisely reflect this. The training process is not "direct" in the sense of a simple regression from $x_0$ to $x_1$; it is a consistency-based method with a specific, cleverly derived target.

- Although the recursive update rule (Equation 13) is intuitively derived from the finite difference, there is no rigorous theoretical connection to the basic ODE. The article would be greatly enhanced by a discussion of how this discrete update rule relates to the numerical integration of the probability flow ODE. For example, under what conditions does the trained $w_\nu(x, 1)$ provide a valid sample from the target distribution?

**Questions:**

- Could you provide a more rigorous justification for why the proposed recursive loss (Eq. 16) leads to a $w_\nu(x_0, 1)$ that generates a sample from the true data distribution $\mu_1$? How does this connect to the theory of flow matching and the probability flow ODE?

- How do your one-step FID scores compare to those achieved by modern distillation-based methods like ADD [19] or DMD [26] on the same datasets and with a similar backbone (DiT-B)? This would help situate your contribution in the broader landscape.

- Please detail the architectural relationship between $v_\theta$ and $w_\nu$. Are they entirely separate networks, or do they share parameters? What is the total parameter count and training time (e.g., GPU days) relative to training a standard Flow Matching baseline?

---

### Note · Authors · 2025-11-14

I have read and agree with the venue's withdrawal policy on behalf of myself and my co-authors.